# ASP-Based Declarative Reasoning in Data-Intensive Enterprise and IoT Applications

**Francesco Calimeri** [1,2] , **Nicola Leone** [1] , **Giovanni Melissari** [2] , **Francesco Pacenza** [1] , **Simona Perri** [1] , **Kristian Reale** [1,2,*] , **Francesco Ricca** [1] and **Jessica Zangari** [1]

1    Department of Mathematics and Computer Science, University of Calabria, 87036 Rende, Italy
2    DLVSystem L.T.D., Via della Resistenza 19/C, 87036 Rende, Italy
\*    Correspondence: kristian.reale@unical.it

**Abstract:** In the last few years, we have witnessed the spread of computing devices getting smaller and smaller (e.g., Smartphones, Smart Devices, Raspberry, etc.), and the production and availability of data getting bigger and bigger. This work presents DLV-EE, a framework based on Answer Set Programming (ASP) for performing declarative reasoning tasks over data-intensive, distributed applications. It relies on the DLV2 system and it features interoperability means for dealing with Big-Data over modern industry-level databases (relational and NoSQL). Furthermore, the work introduces DLV-IoT, an ASP system compatible with "mobile" technologies for enabling advanced reasoning capabilities on smart/IoT devices; eventually, DLV-EE and DLV-IoT via some real-world applications are illustrated as well.

**Keywords:** Answer Set Programming; Non-monotonic Reasoning; DLV; Big Data; SQL; NoSQL; IoT; Raspberry; Android



## 1. Introduction

Answer Set Programming (ASP) [1,2] is an expressive and versatile [3,4] logic programming paradigm that was introduced in the field of non-monotonic reasoning. It allows for defining complex computational problems in a clear and fully declarative fashion. With ASP, a problem can be expressed via a rule-based logic program, whose intended models, called answer sets, correspond one-to-one to solutions. These solutions can be found using ASP systems [5–7].

The intrinsic declarative nature of ASP, combined with its high expressive power, fostered the development of various supporting systems within the scientific community over time [8,9]. This, in turn, has spurred the growth of numerous applications across various fields, including Scheduling [10,11], Workflows [12], Optimization [13], and many others [14]. The availability of robust and reliable ASP systems has been a key factor in this growth.

The DLV system [7] has been one of the first solid and reliable integrated ASP systems. Its project started a few years after the first definition of Answer Set semantics [1,2]. Since its first versions, it has been a suitable tool for applications in academic and real-world scenarios, and significantly contributed both to spreading the usage of ASP and to fostering AI-based technological transfer activities. After years of incremental updates, a brand new version has been released, namely DLV2 [7], a modern ASP system featuring efficient evaluation techniques, proper development tools, versatility, and interoperability. In addition to the standard ASP language, DLV2 offers constructs and tools for further enhancing usability in real-world contexts [15].

The increased use of ASP in the industry has led to the development of advanced libraries [16] and tools [17,18] for supporting programmers, knowledge engineers, and organizations in handling complex projects within real-world domains. However, practical

application scenarios have undergone continuous evolution over the years. Applications requiring reasoning over a large amount of data, possibly varying over time, have been arising and proper solutions have been conceived, e.g., in the context of big data platforms [19], social network analysis [20], biological networks [21], traffic analytic [22]. Moreover, in the latest years, we observed the emergence of compact computing devices and the increasing generation and accessibility of heterogeneous data, e.g., in the context of digital forensics [23], smart cities [24], and activity recognition [25].

This paper presents a framework for developing ASP-based applications in scalable data-intensive, called *DLV Enterprise Edition* (DLV-EE). The framework relies on DLV2, whose existing functionalities have been significantly extended to optimize the evaluation techniques in data-intensive environments. Moreover, the system has been made capable of inter-operate with relational DBMS and NoSQL technologies. Furthermore, DLV-EE provides a service interface, based on the REST philosophy, allowing client environments to interact with it. In addition, the paper introduces DLV-IoT, an ASP system able to shift the execution of reasoning tasks directly on Smart Devices, hence compatible with "mobile" technologies (Android-based devices and Raspberry). This allows edge devices to perform local reasoning over, for example, data provided by their own sensors.

This work also presents an Integrated Development Environment (IDE) consisting of an extended version of the most comprehensive IDE for ASP, namely *ASPIDE* [17]. In particular, the IDE can support the entire life-cycle of ASP development, from program editing to application deployment, combining a cutting-edge editing tool with a collection of user-friendly graphical tools for program composition, debugging, testing, profiling, DBMS access, solver execution configuration, and output-handling.

The rest of the paper is structured as follows. Section 2 presents the DLV-EE framework, with a specific focus on its interoperability means, while Section 3 introduces the DLV-IoT system. Section 3.1 introduces the development tools to facilitate the design, implementation, and deployment of ASP-based applications in the real-world contexts described above. Section 4 illustrates the potential of the proposals via some use cases in the smart city and touristic domains. Eventually, Section 5 draws the conclusions.

## 2. The DLV-EE Framework

This Section presents the framework DLV-EE that empowers the DLV2 system with advanced interoperability means for dealing with Big Data over modern, industry-level, databases.

DLV-EE consists of different modules (see Figure 1) that work together for making the development of effective ASP-based data-intensive solutions viable.

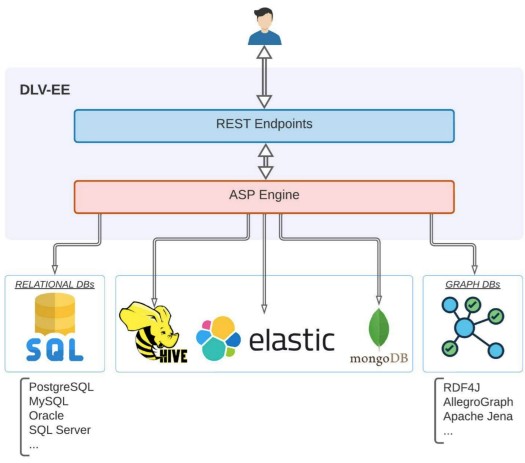

**Figure 1.** DLV-EE Architecture.

In particular, the system emerges from the integration of different versions of DLV2 featuring all the most recent and advanced features, interacting with external database systems (both relational and *NoSQL*, even big-data oriented).

In particular, the system is endowed with proper modules for interacting with external data sources and hence performing reasoning tasks on Relational Database and Graph Database systems via SQL and *SPARQL* [15], respectively. To this aim, specific features have been traditionally integrated into different DLV versions for querying external systems via SQL and SPARQL; such querying is achieved via specific system directives that can be used from within an ASP program. In addition, the support for a set of very popular, industry-standard data management systems has been added, via express modules for *Apache Hive* https://hive.apache.org (and hence the *Apache Hadoop* framework https://hadoop.apache.org) and the *document* database systems *Elasticsearch* https://www.elastic.co/elasticsearch and *MongoDB* https://www.mongodb.com all website accessed on 16 February 2023. It was made possible thanks to the capability of DLV2 of interacting with external sources of computations via *external atoms* [15].

Roughly, proper external atoms are defined along with corresponding handlers (implemented in Python) that are in charge of interacting with the data sources. External atoms are special atoms that can appear only in the rule bodies, whose semantics is provided externally (i.e., from the outside of the logic program) via Python functions. In particular, an external atom has the form (1):

$$\texttt{\&p}(i_1, \ldots, i_n; \ o_1, \ldots, o_m) \quad (1)$$

where &p is an external predicate and $i_0, \ldots, i_n$ and $o_0, \ldots, o_m$ $(n, m \geq 0)$ are input and output terms, respectively. Basically, for each external predicate &$p$ featuring $n/m$ input/output terms, the user must define a Python function whose name is $p$ and has $n/m$ input/output parameters. The function has to be compliant with Python https://docs.python.org/3 version 3. As an example, the following rule $r$ makes use of an external atom with two input and one output terms to concatenate two strings:

```
r: append(X,Y,Z) :- string(X), string(Y), &append_string(X,Y;Z).
```

The program above has to be coupled with the implementation of a Python function, called `append_string`, whose implementation could be as follows:

```
def append_string(X,Y):
    return str(X)+str(Y)
```

External atoms can be both functional and relational, i.e., they can return a single tuple or a set of tuples, as output.

Furthermore, the DLV-EE system has been designed to foster the incorporation of its reasoning capabilities into any other application. This is obtained via the REST endpoint module, which exposes a REST interface. Any Client can use this interface for asking specific ASP programs to be executed; programs are passed via ad-hoc REST calls. A service will then execute the reasoning engine by automatically exploiting the correct module(s) (Hive, MongoDB, Elasticsearch), computing the results, and sending them back to the client in response to the service call. Both the input ASP programs and the returned results will be passed over the network serialized in the JSON format https://www.json.org/.

### 2.1. Reasoning over Hive

*Hive* https://hive.apache.org is a system allowing to build of distributed, large-scale data warehouses featuring a metadata repository for advanced analyses. Hive is built on top of Apache Hadoop https://hadoop.apache.org, which is a robust and very popular open-source framework conceived to facilitate the development of distributed computing over large amounts of data. Hadoop relies on HDFS (Hadoop Distributed File System) https://hadoop.apache.org/docs/r1.2.1/hdfs_design.html for data storage and makes use of *MapReduce* [26] or *YARN* https://hadoop.apache.org/docs/stable/hadoop-yarn/hadoop-yarn-site/YARN.html algorithms to allow the distributed execution across clusters of machines. Hadoop is widely used in industry, as it is highly reliable (e.g., it facilitates the replacement of cluster nodes in case of failure) and scalable (e.g., the computational capacity

can be increased or decreased by just adding or removing nodes). Queries to Hive are given via *HiveQL* https://cwiki.apache.org/confluence/display/Hive/LanguageManual, a language based on SQL. Roughly, when an HiveQL query is issued, Hadoop first analyzes the query and generates an execution plan, then a proper MapReduce or YARN algorithm is used to determine the query answer(s).

A properly designed integration with such a system can allow us not only to retrieve data but also to delegate to the external large-scale database some of the most data-intensive parts of the evaluation. In particular, specific sub-programs of the input ASP program (i.e., those falling in a fragment of ASP corresponding to Datalog) can be evaluated by Hive, without the need of loading the whole input data in the main memory. This can be performed by identifying an evaluation order for the Datalog rules, translating such rules into SQL instructions, and asking Hive for evaluating the so-defined SQL queries. DLV2 then takes the results and evaluates the remaining part of the program, i.e., the part falling in fragments more expressive than Datalog and thus, impossible to be translated into standard SQL.

The following example simulates a potential use case. Suppose that a social network would like to offer its users suggestions about people to invite for a dinner. Besides direct friends, suggestions could also include friends of friends. It is reasonable to assume that close friends could be preferred in suggestions and that the social network can estimate the degree of likeness/unlikeness among subscribed people on the basis of their interests, interactions, and so on. The following ASP program can be adapted to explore (directly and indirectly) the friendship network of a user, namely Giovanni, and then to provide him with suitable suggestions on people that he could invite.

```
f₁: nfriends(10).
f₂: averageAge(25).
r₁: possible_friend(Y) :- close_friend(giovanni,Y ).
r₂: possible_friend(X) :- possible_friend(Y), close_friend(Y,X).
r₃: suggested_friend(Y,A) :- possible_friend(Y), person(Y,A), A>18.
r₄: invite(X) | -invite(X) :- suggested_friend(X).
r₅: :- #count{X: invite(X)} > N, nfriends(N).
r₆: :- #sum{A,X: suggested_friend(X, A),invite(X)} < AVG*N,
        nfriends(N), averageAge(AVG).
r₇: :~ invite(X), suggested_friend(X,_), unlike(giovanni,X,D). [D@1,X]
```

Rules $r_1$ and $r_2$ compute the transitive closure of the friendship relation of Giovanni, restricting the computation to close friends only. Rule $r_3$ suggests a person $Y$ if there is a possible friend and is older than 18. Rule $r_4$ guesses if a suggested friend should be invited or not. Assuming that the number of invited people should be limited, rule $r_5$ ensures that the number of invited friends is not greater than the desired maximum one. Rule $r_6$ imposes that the average age of the invited friends is not smaller than a given value. Rule $r_7$ expresses that preferred solutions are those in which the total degree of dislike among Giovanni and invited people is minimized.

Intuitively, computing the transitive closure of the friendship relation in a social network could be very expensive when performed on a huge database. Thus, traditional main memory ASP systems would struggle in handling it; even the naive import of the friendship relation is not feasible in practice. Hence, given that this subprogram entirely falls into Datalog, its computation can be delegated to Hive, letting the DLV2 be in charge of evaluating the remaining subsequent disjunctive part of the program.

From a practical point of view, the delegation mentioned above is possible thanks to an external atom, namely &bigasp, having the form (2):

```
&bigasp(rules, input, output, db, host:port, user, password; term[, term]). (2)
```

where:

> *rules* a string specifying some Datalog rules whose evaluation is delegated externally on the DB;
>
> *input* a string (possibly empty) featuring a set of ASP facts to be exported to the DB;

*output*  a string specifying the name and the arity of the predicate corresponding to the resulting output relation;

*db*  a string specifying the name of the ODBC DSN (Data Source Name);

*host*:*port* a string reporting the address and the port of the Hive server;

*user*  a string specifying the username willing to connect to the server;

*password* a string specifying the user's password;

*term*  one or more (optional) output terms.

The Python function defining the behavior of such external atom connects to the specified Hive server via the given username and password, exports the given facts (if any), converts the given rules into one or more HiveQL queries, executes such queries on the DB and returns as output a set of tuples that will populate the specified output relation. The output terms correspond to selected fields of the resulting HiveQL queries.

Let us consider again our running example and illustrate how the external atom is intended to be used to externally delegate the computation of suggested friends.

```
r8: suggested_friend(Y,A) :- &bigasp(
        "possible_friend(Y) :- close_friend(giovanni, Y).
        possible_friend(X) :- possible_friend(Y), close_friend(Y,X).
        suggested_friend(Y,A) :- possible_friend(Y), person(Y,A), A>18.", "",
        "suggested_friend/2", "my_db", "192.168.1.1:10000", "my_username",
        "my_password"; Y,A,B,C).
```

Apart from connection parameters and the name of the output relation, the external atom, presented in rule $r_8$, takes as input the extracted rules, invokes the machinery for enabling their evaluation on the external my_db and returns the results as a sequence of tuples representing the suggested friends with their age. Such tuples populate the extension of the suggested_friend relation so that the traditional ASP evaluation can continue.

The external atom &bigasp translates the extracted rules to the following HiveQL queries:

```
1. possible_friend(Y) :- close_friend(giovanni, Y).

    INSERT INTO possible_friend
        SELECT DISTINCT t0.name1 as name,
            '2023-01-11 01:21:09' AS currentStartDate,
            '0' AS recursionLevel
        FROM close_friend AS t0
        WHERE t0.name0='giovanni'

2. possible_friend(X) :- possible_friend(Y), close_friend(Y, X).

    INSERT INTO possible_friend
        SELECT DISTINCT t1.name1 as name,
            '2023-01-11 01:21:09' AS currentStartDate,
            '1' AS recursionLevel
        FROM possible_friend AS t0, close_friend AS t1
        WHERE t0.currentStartDate = '2023-01-11 01:21:09'
            AND t1.currentStartDate = '2023-01-11 01:21:09'
            AND t0.name0=t1.name0 AND t0.recursionLevel= '0'

3. suggested_friend(Y,A) :- possible_friend(Y), person(Y,A), A>18.

    INSERT INTO suggested_friend
        SELECT DISTINCT t0.name0 as name, t1.age as age,
            '2023-01-11 01:21:09' AS currentStartDate
        FROM possible_friend AS t0,person AS t1
        WHERE t0.executiontime = '2023-01-11 01:21:09'
            AND t1.executiontime = '2023-01-11 01:21:09'
            AND t0.name=t1.name
            AND t1.age > 18
```

The translation 1 inserts, in the table *possible_friend*, all close friends of *giovanni*. Considering that *possible_friend* needs to be the result of a recursion, the attribute *currentStartDate*, indicating the moment in which the recursion started, and *recursionLevel*, indicating which

step of the execution was reached, has been introduced. The subsequent recursive executions are made by the translation 2, which performs a join between the *possible_friend* and *close_friend* tables and inserts to *possible_friend* the results of the join where the *recursionLevel* attribute is incremented by 1. With the translation 3, the *suggested_friend* table is populated with the calculated possible friends having an age higher than 18. Finally, the suggested friends are returned to the ASP program and the execution can continue in main memory with the so-filtered data.

### 2.2. Reasoning over MongoDB or Elasticsearch

MongoDB https://www.mongodb.com is a very popular open-source cross-platform document-oriented NoSQL database system, developed in C++. It handles BSON (Binary JSON) https://www.mongodb.com/json-and-bson, a JSON extension that features a broader data type support, and it is used in a wide range of different application domains, including the gaming industry (SEGA, Electronic Arts) and e-commerce (eBay) https://www.mongodb.com/who-uses-mongodb. Thanks to its distributed nature, it can also be used as a distributed file system, supporting the storage of large files and taking advantage of replication and load balancing across multiple servers. Data are stored in structures called "collections", that contain documents without a fixed schema; the supported query language allows to filter and sort data over every field and to perform any kind of aggregation. For instance, with aggregation queries one can group documents from collections that meet specific criteria and apply over them aggregation functions such as count, average, or sum.

To make DLV2 interact with *MongoDB* a specific external atom has been designed. Instances of these external atoms can be featured in some rule bodies; during the evaluation of the program, the corresponding Python module is invoked for fetching the data from the DBMS by querying the system and transforming the results in ASP facts. The facts are in turn used by DLV2 for evaluating the ASP program. The external atom has the form (3):

```
&mongo(host, port, database, collection, query, key, aggr; term[, term]).  (3)
```

where:

| | |
|---|---|
| *host* | is the address of the MongoDB server. |
| *port* | is the port of the MongoDB server. |
| *database* | is the name of the database over which the query has to be performed. |
| *collection* | is the name of the collection from which data is to be retrieved. |
| *query* | is the query to be executed on MongoDB. |
| *key* | is needed to select fields within the document; this parameter determines the arity of the predicate. If an empty string ("") is specified, all fields are returned. |
| *aggr* | this parameter informs the Python module about the presence of aggregate functions, hence providing instructions about the search method to be used: `"yes"` indicates to use db.aggregate(), while `"no"` indicates to use db.find(). |
| *term* | one or more (optional) output terms determining the values of the field(s) retrieved via the query. |

The following example illustrates how an ASP program can make use of data retrieved via MongoDB. Considering the ASP rule $r_1$ below:

```
r₁: b(X,Y) :- &mongo("localhost", 27017, "admin", "football",
        "{age: {$gt: 30}}", "_id: 0, name: 1, surname: 1","yes"; X, Y ).
```

$r_1$ can be used to fill in the extension of the head predicate $b$ having arity 2 with the names and the surnames of the football players appearing in the collection "football" who are over 30 years old.

During the evaluation of the logic program, DLV2 invokes the Python module corresponding to the external atom `mongo` where the input terms contain the information needed for establishing the connection with the MongoDB server and performing the query. DLV2 waits for the results, that return as JSON documents, for instance, $json_1$ and $json_2$:

```
json₁ : { "player": { "name": "Cristiano", "surname": "Ronaldo" } }
json₂ : { "player": { "name": "Leo", "surname": "Messi" } }
```

Such documents are then transformed into tuples that bind the output parameters `X` and `Y`, used by DLV2 for evaluating the rule.

Elasticsearch https://www.elastic.co/elasticsearch is a full-text search engine based on Lucene https://lucene.apache.org, a free open-source API for information extraction, widely used in the development of search engines. It is also a document-oriented DBMS that stores data over a distributed database. Documents are collections of fields represented in JSON format; each document belongs to an index and is identified by a unique key. Load balancing over the database distributed architecture is guaranteed by the division into Shards (partitions).

The interoperability between DLV2 and Elasticsearch is obtained by defining a specific external atom. In this case, the external atom has the form (4):

$$\&\, elastic(\, host,\ port,\ query, index;\ term\ [,term])\ (4)$$

where:

| | |
|---|---|
| *host* | is the address of the Elasticsearch server. |
| *port* | is the listening port of the Elasticsearch server (9200 is the standard one). |
| *query* | is the query to be executed on MongoDB. |
| *index* | is a string representing the index where the query should be performed. |
| *term* | one or more (optional) output terms determining the values of the field(s) retrieved via the query. |

Similarly to the MongoDB case, the external atom of Elasticsearch can be exploited to perform a query, e.g. for obtaining all the football players as in the rule $r_2$ below:

```
r₂ :  b(X,Y) :- &elasticsearch("localhost", 9200,
      "{'_source':['teams','player'],'query':{'match_all':{}}}","anElasticIndex";X,Y).
```

Also in this case the Python module invokes Elasticsearch to pass the query and obtain the results.

## 3. DLV-IoT

The Internet of Things (IoT) is a rapidly growing technology trend that is changing the way people live, work, and interact with the world. It refers to an interconnected network of devices, such as vehicles, home appliances, and other items which are embedded with sensors, specific firmware, and network connectivity, allowing them to collect and exchange vast amounts of data. The use of IoT devices is predicted to expand even more in the future, thanks to the new technologies coming from domotics, home automation, healthcare, and smart cities [27]. Moreover, IoT devices are particularly attractive since they allow us to make decisions, automate processes, and improve efficiency in many different contexts. For the above reasons, enabling advanced and complex reasoning capabilities in IoT applications, today, could be largely beneficial, especially when massive groups of sensors need to be leveraged.

In general, in the IoT context, devices first acquire information about the surrounding environment, then a decision-making logic takes actions based on the input knowledge. Typically, due to the limited availability of computing power or memory on the edge devices, data is sent to remote/cloud systems that are in charge of actually performing the computation. However, data collected can come at a high pace and high volumes, thus suggesting that the ability to reason on such data locally and in real-time could benefit many advanced applications.

*DLV-IoT* is an ASP system conceived to be profitably used in such contexts, as it can shift the execution of reasoning tasks directly on Smart Devices. DLV-IoT consists of a specific version of DLV that analyzes the requested reasoning task and input data, and performs an estimation of the computational workload, thus deciding whether it can be reasonably executed locally; if this is not the case, it interacts with DLV-EE via the REST

services made available via the introduction of a REST endpoint, thus enjoying the full computational power of the DLV-EE system, presumably running over more powerful infrastructures (see Figure 2).

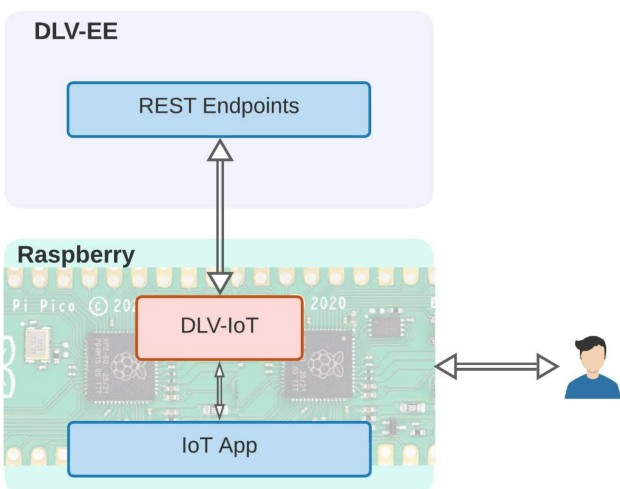

**Figure 2.** Typical architecture of a DLV-IoT-based application over Raspberry.

To date, besides the widespread mobile operating system Android https://www.android.com/, DLV-IoT is also able to run on *Raspberry Pi* https://www.raspberrypi.com/, which is one of the most commonly used systems for collecting, aggregating and analyzing data coming from a network of IoT devices. Raspberry Pi is a tiny, low-cost, single-board computer that has become increasingly popular for IoT applications thanks to its compact size, portability, versatility, and low power consumption. Moreover, Raspberry Pi can be also directly connected to a variety of sensors, allowing it to gather data immediately from the physical world.

The current version of DLV-IoT performs the local evaluation only in the case of Datalog programs; when full ASP language is present, it defaults to DLV-EE; more accurate estimations of the computational workload are under consideration.

*3.1. Development Tools for DLV-EE and DLV-IoT*

The evolution and the introduction of more advanced hardware and software technologies generally go hand in hand with the implementation of proper tools for supporting developers during the entire development cycle of a given application, such as proper Integrated Development Environments (IDEs). As for "standard" ASP, and DLV in particular, *ASPIDE* [17] is one of the most comprehensive IDE for ASP which was followed by further extensions over the years.

To support the development of applications based on DLV-EE and DLV-IoT, i.e., applications that require ASP solvers to go beyond reasoning in main memory and gathering input from local files, this work *ASPIDE* has been properly extended via a synergic integration of multiple Integrated Development Environments suitably adapted to our purpose (Figure 3).

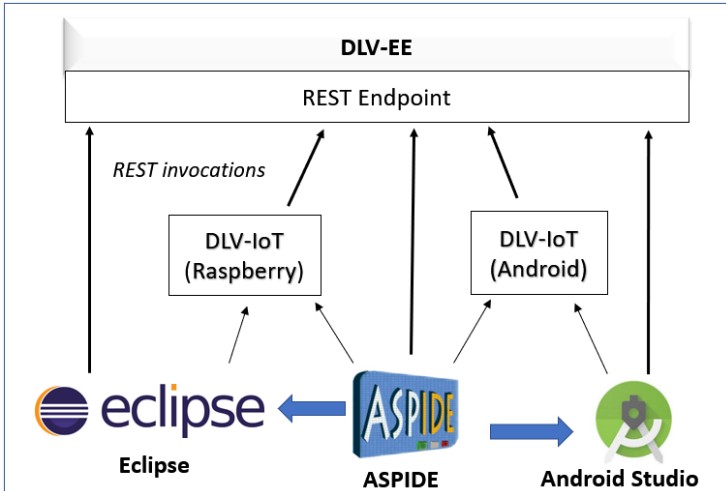

**Figure 3.** Integrated Development Environment for DLV-EE and DLV-IoT.

### 3.1.1. Development of DLV-EE Solutions

In the case of applications that make direct use of DLV-EE, the development can rely on direct access to the REST services; however, one can also use *ASPIDE* which has been purposely extended to ease the whole process. In particular, once the ASP programs are written, it is possible to decide either to save them directly to the local disk or interact with the REST interface to access a remote workspace and store ASP programs and facts directly to the DLV-EE storage location. Moreover, it is possible to invoke DLV-EE from *ASPIDE* itself by exploiting the REST interface.

### 3.1.2. Development of DLV-IoT Solutions

For the development of DLV-IoT based solutions for *Android* and *Raspberry Pi* systems, in this work *ASPIDE* has been extender to make the system suitable for implementing and deploying ASP programs into external libraries. In such a way, the final libraries can be integrated, on one hand, to the *Android Studio* environment https://developer.android.com/studio for the development of software solutions based on *Android*, and on the other hand, to the Eclipse environment https://www.eclipse.org/, for implementing e.g., *Raspberry Pi* solutions. More in detail, an ASP programmer can first write the ASP program in *ASPIDE*, export the program into an AAR https://developer.android.com/studio/projects/android-library/JAR https://docs.oracle.com/javase/8/docs/technotes/guides/jar/jarGuide.html library and then import the library into a Java program for *Android* or *Raspberry Pi* applications: the Java programmer just needs to import the library into the desired environment like *Android Studio* or Eclipse and use it (note that the DLV-IoT system will be included in the library as well).

## 4. DLV-EE and DLV-IoT at Work: Some Use Cases

This section presents two real-world scenarios and shows how applications based on DLV-EE and DLV-IoT can be conveniently developed. The first scenario consists of an *Android* app, namely *NavTour*, allowing users to exploit an intelligent navigator system (*DLVNavigator*); the second scenario makes use of the DLV-IoT system in a Smart Cities context.

### 4.1. Planning Touristic Itineraries

*DLVNavigator* is a web service, based on DLV-EE, which has been endowed with additional features to automatically generate tourist itineraries. The web service can be easily accessed from multiple clients via a *RESTful* architecture that exposes *APIs* capable of providing a rich set of services, which are briefly described in the following. The service provides the user with a planned itinerary enriched with information about locations

and the visiting times of the points of interest (*POIs*) (e.g., historic squares, museums, monuments, etc.). Itineraries are guaranteed to be free of loops and dead ends, also ensuring compliance with the preferences and time constraints coming from the user and paying attention to other details (e.g., do not place *POIs* to be visited in time slots in which they are not available to the public).

To generate a fully customized itinerary, which results as close as possible to desiderata from the user, some profiling functionalities have been implemented; in particular, users can indicate their preferences regarding the type of *POIs* they intend to visit expressed by numerical values between 0 and 10, for each category of points of interest. Registration (and authentication) functions have been made available to the user to associate itineraries and profiles to a specific user account.

The service is made available to mobile users via an Android App called *NavTour* (see Figure 4). When a new tour has been planned by *DLVNavigator*, it is available in the app; all the stages are reported, and the user can start the tour. When the user physically moves from one place to the next one, the app automatically keeps track of the progresses; interestingly, the tour is dynamically managed: in case the user spends more time than what was originally allotted in the schedule, a rescheduling is automatically determined, taking into account the remaining time and user preferences.

For computing the tourist itinerary, *DLVNavigator* relies on a proper ASP program consisting of two layers. The first layer is in charge of selecting a number of *POIs* complying with some given constraints (i.e., number, duration, etc.). The selection of *POIs* is optimized according to user preferences (e.g., how many *POIs* for each category) and total costs (e.g., ticket prices). The second layer builds the tour by defining the sequence of *POIs* to visit, taking into account entry/exit times and the available time budget; the tour can also be optimized according to other specific desiderata (e.g., minimizing the distances between consecutive *POIs* in the tour). Details about the implementation and the ASP programs employed can be found at https://www.mat.unical.it/ricca/aspide/dlvee.

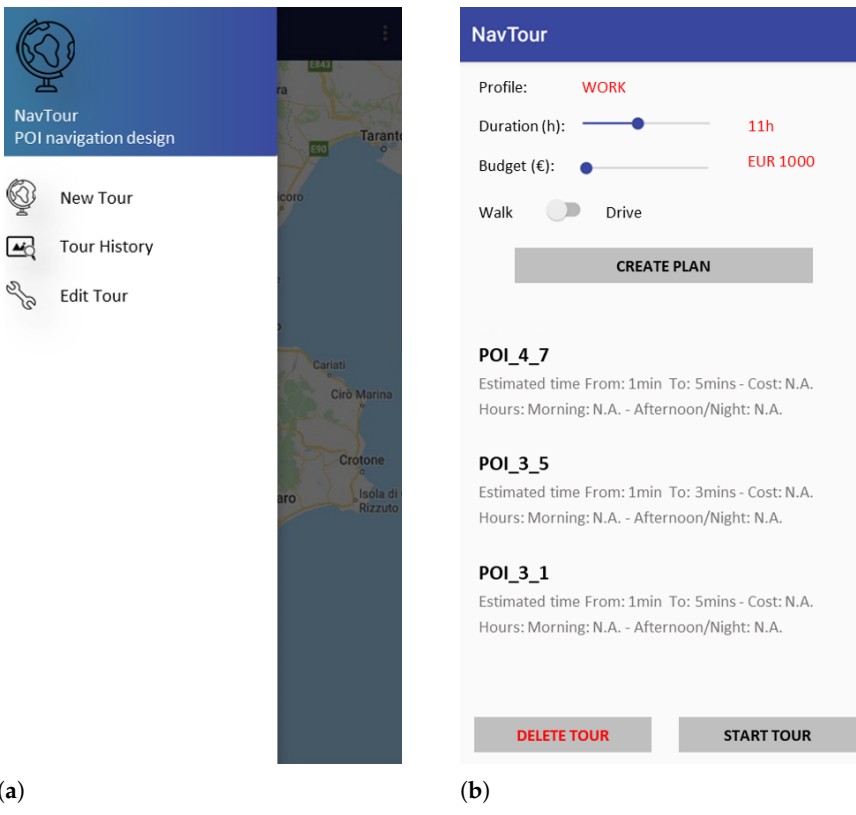

(**a**)                    (**b**)

**Figure 4.** The *NavTour* Android application. (**a**) New itinerary; (**b**) Planned Tour.

### 4.2. Controlling Traffic Lights in Road Crossings

In a smart city [28? ], digital technologies, such as edge devices or sensors, are scattered throughout the city to collect data. Leveraging technological solutions from different areas, such as IoT, Social Computing, and Artificial Intelligence, new information about the surrounding environment is inferred from collected data [30,31]. Such information is used for automatizing a "smart" management of infrastructures, resources, and services.

The following part illustrates how DLV-IoT can be used as a smart city technology.

As a target use case, consider a simplified setting consisting of three roads with two crossings, namely c1 and c2, as shown in Figure 5. On each crossing, there are four traffic lights, one per each traveling direction, i.e., *s1, s2, s3*, and *s4*. For the sake of simplicity, it is supposed that vehicles can only go straight without turning right or left and that the behavior of the two crossings follows the rules listed below.

- In each crossing, the two traffic lights on the same road must simultaneously have the same status; thus, they are grouped in the pairs $<s1, s4>$ and $<s2, s3>$. In addition, when $<s1, s4>$ have the red status, $<s2, s3>$ have the green status and vice-versa. Green means that vehicles can pass; red means that vehicles have to stop.
- Every 10 s, the traffic lights switch their status.
- In case a pedestrian wants to request to cross at a traffic light, it and its paired traffic light turn red. Consequently, the status of both traffic lights in the other pair of the same crossing switches to green.
- The *green wave* can be enabled at certain times of the day, to force the pair $<s1, s4>$ of both crossings *c1* and *c2* to become simultaneously green, while all the other traffic lights must turn red.

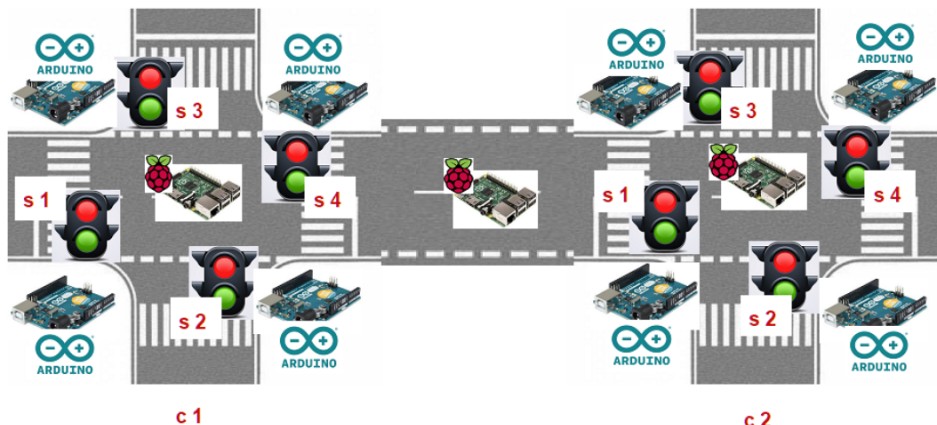

**Figure 5.** A Smart City use case: crossroads with traffic lights.

This scenario was simulated using some *Arduino* and *Raspberry Pi* devices. The configuration is depicted in Figure 5: each traffic light is emulated with an *Arduino* https://www.arduino.cc/ device and each crossing is equipped with a *Raspberry Pi* device; moreover, a further central *Raspberry Pi* device coordinates the traffic lights in green wave situations.

ASP can be profitably used to model the functioning of the crossings using DLV-IoT as a reasoning engine, executed on the *Raspberry Pi* devices. In particular, for each crossing, the corresponding *Raspberry Pi* device executes DLV-IoT over an ASP program for defining how to control the local functioning of its traffic lights. The ASP program receives as input some facts modeling the topology of the corresponding crossing. For instance, for *c1*, the following facts, from $f_1$ to $f_5$, states that *c1* is a crossing and that *s1, s2, s3, s4* are traffic lights related to *c1*.

$f_1$: `crossing(c1).`
$f_2$: `traffic_light(s1,c1).`
$f_3$: `traffic_light(s2,c1).`

```
f₄: traffic_light(s3,c1).
f₅: traffic_light(s4,c1).
```

Similar facts, in which *c1* is replaced by *c2*, are the input of the ASP program controlling the crossing *c2*. The ASP program, modeling the behavior of each crossing, is composed as follows:

```
r₁: samePair(s1,s4,C) :- crossing(C).
r₂: samePair(s2,s3,C) :- crossing(C).
r₃: status(T,C,red) | status(T,C,green) :- traffic_light(T,C).
r₄: :- status(T1,C,Status1), status(T2,C,Status2),
       samePair(T1,T2,C), Status1!=Status2.
r₅: :- status(T1,C,Status), status(T2,C,Status),
       T1!=T2, not samePair(T1,T2,C).
```

Rules $r_1$ and $r_2$ define the grouping in pairs of traffic lights on the same road. Rule $r_3$ guesses the status of each traffic light. Rules $r_4$ and $r_5$ are strong constraints allowing to discard solutions in which two traffic lights of the same road have different statuses (i.e., $r_4$) and two different traffic lights not part of the same pair have the same status (i.e., $r_5$). The *Raspberry Pi* of each crossing generates a first status assignment for each traffic light and every 10 seconds the statuses are automatically turned. In case a pedestrian requires to cross a road, e.g., at the traffic light *s1* of *c1*, the *Raspberry Pi* of *c1* is notified, and consequently, it re-executes DLV-IoT over the same ASP program above with the addition of the fact `status(s1,c1,red)`. The statuses of the traffic lights are accordingly updated and then, continue to be turned every 10 seconds. The central *Raspberry Pi* is in charge of synchronizing both crossings in case of a green wave. To this end, DLV-IoT is executed over the same program above with the addition of the strong constraint reported below (rule $r_6$).

```
r₆: :- greenwave, state(s1,C,red).
```

Hence, when a green wave is requested (i.e., a fact `greenwave` is given as input), the status of the traffic lights *s1* of both *c1* and *c2* has to be green. The constraint $r_4$ ensures that the traffic lights *s4* of both *c1* and *c2* are green as well. The central *Raspberry Pi* can thus directly update the statuses of all traffic lights overriding the default behavior.

## 5. Conclusion and Discussion

This paper presented DLV-EE, an ASP-based Framework for performing declarative-based reasoning tasks over Big Data, classical relational database systems, and NoSQL databases. Moreover, DLV-IoT, a variant of DLV2 geared towards IoT technologies, was proposed for easing the implementation of ASP-based mobile and distributed applications. The specific development tools, designed for DLV-EE and DLV-IoT, and based on *ASPIDE* were also discussed. Finally, the potential of the proposals was highlighted with the help of some use cases. DLV-EE, DLV-IoT and the related development tools can be downloaded at https://www.mat.unical.it/ricca/aspide/dlvee.

As far as future work is concerned, extensions are planned for both the DLV-EE framework and the development tools, with the aim of dealing with more data sources, both relational and NoSQL, and further improving performance in data-intensive contexts. Furthermore, compatibility of DLV-IoT is planned to be improved beyond the currently supported Raspberry and Android architectures; also, improvements are programmed on the workload estimator, for more fine-tuned decisions aimed at minimizing the need for computation carried away from the edge devices.

**Author Contributions:** Conceptualization, F.C., N.L., S.P. and J.Z.; Methodology, F.C., N.L., F.P., S.P., F.R. and J.Z.; Software, F.C., G.M., K.R., F.R. and J.Z.; Validation, F.C., N.L., S.P. and F.R.; Formal analysis, F.C., F.P., S.P., F.R. and J.Z.; Investigation, F.C., K.R., F.R. and J.Z.; Resources, G.M., F.P., K.R. and J.Z.; Data curation, N.L. and G.M.; Writing—original draft, F.P., S.P. and J.Z.; Writing—review & editing, F.C., F.P., S.P., F.R. and J.Z.; Visualization, F.C., G.M., S.P. and F.R.; Supervision, N.L., S.P. and F.R.; Project administration, F.C. and N.L.; Funding acquisition, N.L. All authors have read and agreed to the published version of the manuscript.

**Funding:** This research received no external funding.

**Institutional Review Board Statement:** Not applicable.

**Informed Consent Statement:** Not applicable.

**Data Availability Statement:** Not applicable.

**Acknowledgments:** This work has been partially supported by : (i) POR CALABRIA FESR-FSE 2014-2020, project "DLV Large Scale: un sistema per applicazioni di Intelligenza Artificiale in architetture data-intensive e mobile", CUP J28C17000220006; (ii) PRIN PE6, Title: "Declarative Reasoning over Streams", funded by the Italian Ministero dell'Università, dell'Istruzione e della Ricerca (MIUR), CUP:H24I17000080001; (iii) PON-MISE MAP4ID, Title: "Multipurpose Analytics Platform 4 IndustrialData", funded by the Italian Ministero dello Sviluppo Economico (MISE), CUP: B21B19000650008; (iv) PON-MISE S2BDW, Title: "Smarter Solution in the Big Data World", funded by the Italian Ministero dello Sviluppo Economico (MISE), CUP: B28I17000250008. Also, this work contributes to the basic research activities of the WP9.1: "KRR Frameworks for Green-aware AI" supported by the PNRR project FAIR - Future AI Research (PE00000013), Spoke 9 - Green-aware AI, under the NRRP MUR program funded by the NextGenerationEU.

**Conflicts of Interest:** The authors declare no conflict of interest.

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
