# Peer review of "ASP-Based Declarative Reasoning in Data-Intensive Enterprise and IoT Applications"

_algorithms, doi:10.3390/a16030159_

Round 1

Reviewer 1 Report

This paper surveys authors' recent development of a set of software systems based on answer set programming, which are derived from the DLV system.

As major points, the paper provide the latest information about industrial and social applications of DLV systems in the real world, which will be interesting to  the audience of the special issue.

As minor points, this paper has less mathematical and computational explanation about the techniques used in the systems. However, this weakness comes from the nature of this paper since it aims to provide a glimpse of wide applications of the ASP. 

Overall, I enjoyed this paper, and I would like to see it in the special issue. 

Detailed comments to the authors (necessary to revise): 

page 1, line 20: "the development of various supporting systems within the scientific community over time" 
=> Append at least one reference as the evidence to support this claim. 

page 1, line 26: "DLV"
=> "The DLV systems"? (I leave this decision to the authors since I am not sure about it.

page 2, "SparQL" at line 78 and "SparQL [23]" at line 60
=> Move the citation "[23]" to the first occurence of this keyword at line 78.

page 10, Section 4.1, the first paragraph from line 391 to line 421
=> This paragraph is too long to capture what the authors would like to explain. Please divide this paragraph into a few of shorter ones to improve the readability. Maybe, you can split it at line 399 and line 411. 

Overall) For all softwares, systems, and standards which the reader may not has sufficient knowledge, please put the appropriate URL (or citation) for them. Although the many of them are correctly handled, some of them lack sufficient information. I list them below: 

page 2,
- line 36: "the ASP-COre-2 standard language"
- line 45: DLV-EE
page 4
- line 113: "JSON format"
- line 118: "HDFS"
- line 119: "MapReduce" and "YARN"
- line 123: "HiveQL"
page 6: 
- line 246: "BSON"
page 8: 
- line 341 "Raspberry Pi"
page 9: 
- line 374: Android
page 10: 
- line 380: "AAR/JAR library"
- line 382: "Eclipse"
- line 390: "DLVNavigator"
page 11: 
- line 446: "Arduino"
===

Author Response

Dear Reviewer,

Please see the attachment for our response to the review.

Kind Regards,

Kristian Reale

Reviewer 2 Report

This study deployed an Answer Set Programming (ASP) framework for performing declarative reasoning tasks in the Data-Intensive and Distributed applications based on the DLV-IoT system. The current work served the aim of the Algorithm Journal. However, the methodology needs to be clarified, and the results need in-depth explanations and validation. Also, the novelty and significance of the manuscript must be presented clearly. Moreover, an in-depth analysis of the results and how the suggested method is constructed and evaluated must be conducted. The introduction must present a detailed background of the current problem and solutions. Also, it should clearly explain the new findings in this work and the research gaps. Besides, the technical writing has many grammatical errors, making it hard to read. The authors should justify the new contribution of the proposed work as similar studies have been carried out in the existing literature.  

In addition to the following:

- The title did not reflect the proposed system's objectives and meaning.

-Enhance the abstract and conclusion to present the current problem and findings. 

- Describe and conclude the advantages and disadvantages of each mentioned method in the literature studies.

-It needs more comprehensive evaluations and comparisons with other researchers (mentioned in the related work) to validate the obtained results supported by graphical and tabular data.

-Please add and use the recent references 2022,20221.

-What are the current research gaps in the studies mentioned in the literature survey, and how will this work fill them?

-The research paper should be written in the third person's perspective; words such as "we", "our," etc., must be avoided.

- Avoid using many references together, such as L13: [1-3], L18: [5-8], L24: [9-17], L30 [18-20], L40: [24-28], etc. You should classify the studies and write a proper paragraph bout each study or category.

- Add a reference to any figure you used for others, such as Fig.1, Fig.2, etc.

- Give numbers for the equations, such as L92, L99, etc.

- Assemble lines 145-153 into one Figure (Algorithm).

- Assemble lines 190-195 into one Figure (Algorithm). Do the same for any coding statements.

-Too-long sentences make the meaning unclear. Consider breaking it into multiple sentences—for example, L72-L30; L33-L35; L44-L47; etc.

-Many grammatical or spelling errors make the meaning unclear, and sentence construction errors need proofreading. Improve the English language, redaction, and punctuation in general. The manuscript should undergo editing before being submitted again.

The following are some examples:

L1: have been witnessing the  ……It should be….    we have witnessed

L3: In this work we present……It should be….    This work presents

L5: system, and features ……It should be….    system and features 

L6: industry-level, databases (both relational and NoSQL).   ……It should be….    industry-level databases (relational and NoSQL).

L38: in industry has led   ……It should be….    in the industry has led

L41: in the latest years we  ……It should be….    in the latest years, we

L44: In this paper, we present  ……It should be….          This paper presents

L44: for the development of ASP-based ……It should be….      for developing ASP-based

L48: been made capable to inter-operate with both……It should be….      been capable of inter-operating with

Author Response

(The authors gave the same response as above.)

Round 2

Reviewer 2 Report

The author did excellent work in addressing my comments. The revised manuscript is improved to the level that could publish in the current form with minor corrections based on the editorial board's opinion.

-The research paper should be written in the third person's perspective;

words such as "we", "our," etc., must be avoided.

- Avoid using many references together, such as [1-3], [5-5], [10–14], [16–18], etc. You should classify the studies and write a proper paragraph bout each study or category.

-To make your text more readable, you need to renumber the in-text references to be an accending form ( 1, 2, 3, 4, ...).

- Add a reference to any figure you used for others, such as Fig.1,

Fig.2, etc.

- Give numbers for the equations, such as L93, L100, etc.

- Assemble lines 146-154 into one Figure (Algorithm).

- Assemble lines 191-196 into one Figure (Algorithm). Do the same

for any coding statements.

-Many grammatical or spelling errors make the meaning unclear, and

sentence construction errors need proofreading. Improve the English

language, redaction, and punctuation in general. The manuscript

should undergo editing before being submitted again.

The following are some examples:

L1: have been witnessing the ......It should be.... we have witnessed

L3: In this work we present......It should be.... This work presents

L41: in the latest years we ......It should be.... in the latest years, we

L46: In this paper, we present ......It should be.... This paper presents

L48: been made capable to inter-operate with both......It should be.... been capable of inter-operating with

L55: consisting on an extended  .....It should be....  consisting of an extended 

L61: In Section 2 we present  .....It should be....  In Section 2, we present

L63: aimed at facilitating design  .....It should be.... to facilitate the design

L64: In Section 4 we illustrate .....It should be....  In Section 4, we illustrate 

Author Response

We would like to sincerely thank the reviewers for the time they spent on our work and for their useful comments and suggestions, which helped us to improve our work in several respects.

We have updated the manuscript accordingly to your last comments.